# Apportionment and Spatial Pattern Analysis of Soil Heavy Metal Pollution Sources Related to Industries of Concern in a County in Southwestern China

**DOI:** 10.3390/ijerph19127421

**Published:** 2022-06-16

**Authors:** Xiaohui Chen, Mei Lei, Shiwen Zhang, Degang Zhang, Guanghui Guo, Xiaofeng Zhao

**Affiliations:** 1Center for Environmental Remediation, Institute of Geographic Sciences and Natural Resources Research, Chinese Academy of Sciences, Beijing 100101, China; luokeangcxh@163.com (X.C.); zdg_biology2@126.com (D.Z.); guogh@igsnrr.ac.cn (G.G.); zhaoxf2011@163.com (X.Z.); 2University of Chinese Academy of Sciences, Beijing 100049, China; 3School of Earth and Environment, Anhui University of Science and Technology, Huainan 232001, China; shwzhang@aust.edu.cn

**Keywords:** heavy metals, industries of concern, source apportionment, spatial patterns, prevention distance

## Abstract

Soil heavy metal pollution is frequent around areas with a high concentration of heavy industry enterprises. The integration of geostatistical and chemometric methods has been used to identify sources and the spatial patterns of soil heavy metals. Taking a county in southwestern China as an example, two subregions were analyzed. Subregion R1 mainly contained nonferrous mining, and subregion R2 was affected by smelting. Two factors (R1F1 and R1F2) associated with industry in R1 were extracted through positive matrix factorization (PMF) to obtain contributions to the soil As (64.62%), Cd (77.77%), Cu (53.10%), Pb (75.76%), Zn (59.59%), and Sb (32.66%); two factors (R2F1 and R2F2) also related to industry in R2 were extracted to obtain contributions to the As (53.35%), Cd (32.99%), Cu (53.10%), Pb (56.08%), Zn (67.61%), and Sb (42.79%). Combined with PMF results, cokriging (CK) was applied, and the z-score and root-mean square error were reduced by 11.04% on average due to the homology of heavy metals. Furthermore, a prevention distance of approximately 1800 m for the industries of concern was proposed based on locally weighted regression (LWR). It is concluded that it is necessary to define subregions for apportionment in area with different industries, and CK and LWR analyses could be used to analyze prevention distance.

## 1. Introduction

With the rapid development of the global economy, industrialization is intensifying, and cities are expanding. Heavy metals in industrial waste have affected the surrounding soil, making it easier for heavy metal concentrations to reach high levels of toxicity [1]. Heavy metal pollutants do not degrade easily, have poor mobility, and easily accumulate. Additionally, mining and smelting operations play a critical role in the accumulation of heavy metals in local soils, especially in areas with resource-dependent economies. For example, the mining of polymetallic ores provides not only the targeted metals but also vast amounts of by-products, such as tin and copper associated with sulfide ore [2]. The areas affected by mining are always seriously polluted by heavy metals, which are emitted from the mining waste and tailings and are transported by runoff and human activities [1,3]. The concern is that the mining sites, tailings, and associated equipment could be pollution sources for a long time [4]. In addition, emissions from smelting may be distributed widely via atmospheric deposition, and this is especially true for metals such as As, Cd, and Pb, which can form into oxides and condense into particulates at high temperature [5].

Identification and quantitation of the sources of pollutants in soils are critical for risk assessments and the development of reclamation strategies. The apportionment of pollution sources began with atmospheric particulates, and the technical system consisted of an emission inventory, mass diffusion model, and receptor model [6,7,8,9]. The main objects of the apportionment for soil pollution sources are heavy metals and some organic matter (e.g., polycyclic aromatic hydrocarbons, PAHs), which are always widespread and do not easily degrade [10,11]. Receptor models have been the primary method because of the uncertainty of emissions inventory information and the complexity of mass diffusion models [9,12,13,14]. Currently, many studies have estimated the spatial patterns of source factors extracted from receptor models to increase the credibility of pollution source identification [13,15]. When using receptor models, few studies have paid attention to spatial variations in the pollution sources, e.g., the areas that have a higher density of key industries that emit pollution.

In China, investigations of the soil environment around enterprises in key industries have been conducted [16]. Due to a literal misunderstanding, the key industries were named as the industries concerned in these studies. At the beginning of the survey, risk screening was carried out through attribute scores of large-scale areas [17,18], but the soil environment around the industries of concern should be investigated before health risk assessments of contaminated sites are conducted to provide suggestions about the spatial patterns of the environmental protection and the priority pollutants on the county level, which is also the target scale in this study. The study area is a remote region with predominantly silver and copper mining in the early stage. However, after the 1840s, it became a strategic area for tin, lead, and zinc mining and smelting. These mineral resources have been gradually exhausted since the 1990s, but historic mining has been inseparable from the local businesses and the livelihoods of the residents, so determination of a sustainable development scheme for this region has become a critical issue. Most environmental surveys and studies of this area have paid attention to the issues of soil heavy metal pollution, which mainly originates from mining and smelting activities [19,20,21]. In addition to tin, multiple minerals occur in the study area, including copper, lead, and zinc ores. Some of the mineral deposits are distributed within the tin mineralization zone, while others are distributed outside of this zone. The characteristics of soil heavy metal pollution in polymetallic mining area can help to distinguish the contribution of major industrial pollution sources and give the protection distance of soil heavy metal pollution of main industrial land in this area through its spatial distribution with enterprise land.

Specifically, the main objectives of this study are as follows: (i) to explore the relationship between the soil heavy metals and the spatial distribution of the pollution sources from industries of concern; (ii) to extract the factors of the soil heavy metals through positive matrix factorization (PMF) to estimate the contributions of the industries of concern; and (iii) to determine the spatial patterns of the heavy metals originating from certain factors through cokriging (CK), which makes use of the homology of heavy metals to improve the accuracy of the assessment, so as to determine the variations in the heavy metals with distance from the nearest source through locally weighted regression (LWR).

## 2. Materials and Methods

### 2.1. Study Area and Investigation of Pollution Sources

The study area (102°54′–103°25′ E; 23°01′–23°36′ N) contained more than 400 key enterprises and consists of five towns, with a total area of 992.05 km^2^, which are in a supergiant tin polymetallic district located along the suture zone of the Indian and Eurasian plates on the southwestern edge of the China sub-plate. This area has a subtropical mountain monsoon climate, with an annual average temperature of 15.9 °C and an annual rainfall of 1292.8 mm. The parent materials of the area are very complex, which is mainly composed of limestone and minor dolomite, and the main soil types are yellow-brown and red soils.

Using a combination of remote sensing and historical business information, a field survey of the pollution sources related to the nonferrous metal industry was conducted, and five types of sources were confirmed and vectorized, which provided their spatial positions and areas (Appendix A). The results of the field survey revealed that 3000 ha of land may have been the source of the soil pollution in subregion 1 (R1). Mining had led to destruction of the landscape, and tailings (i.e., large particles that settle at the bottom of the flotation tank), which are of lower economic value, were directly discarded in a natural depression during a time when environmental protection was not of great concern. Additionally, in subregion 2 (R2), 935 ha of land could be a potential source of soil pollution, and this land was mainly covered by slag produced by lead and zinc smelting.

### 2.2. Sample Collection and Chemical Analysis

The collection and analysis of 230 samples were completed in 2018, and the database used in this study also incorporated the results of geochemical surveys conducted in 2013 and 2015 (390 samples). In R1 and R2, 389 and 231 samples were collected and analyzed, respectively. All surveys were conducted using the same sampling and geochemical analysis methods, and the soil samples were collected according to the distribution of the different land use (dry land, *n* = 412; paddy land, *n* = 123; garden plot, *n* = 85). Each composite sample (1 kg) was composed of five subsamples collected at the central point and four additional points within an area of 5 m^2^. The 3S technique was used for the sampling conducted around the mining area, industrial plants, mining waste heaps, smelting slag heaps, and tailings ponds. In addition, several samples were collected away from the pollution sources to study the pollution from atmospheric deposition and surface runoff. Moreover, the sampling density around the sources was four samples per square kilometer (Figure 1).

Upon receipt, the samples were dried in a lyophilizer and sieved (2 mm mesh), and then, the stones, litter, and roots were removed. The total major element contents (K, Ca, Mn, and Fe) of the samples were analyzed using an X-ray fluorescence spectrometer (Niton FXL analyzer, Thermo-Fisher Scientific, Waltham, MA, USA) [22]. Then, the samples were digested in HNO_3_ and H_2_O_2_ using method 3050B (USEPA, 1996). The total As concentration was analyzed using atomic fluorescence spectroscopy (AFS-9800, Haiguang Instrument Co., Beijing, China), and the Cd concentration was analyzed using graphite furnace atomic absorption spectrometry (contrAA700, Analytikjena, Jena, Germany). The other minor elements were measured using inductively coupled plasma optical emission spectrometry (Optima 5300DV, PerkinElmer, Boston, MA, USA). The detection limits of As, Cd, Cu, Cr, Ni, Pb, Sb, and Zn were 0.10, 0.05, 0.10, 0.10, 0.05, 0.10, 0.05, and 0.50 mg/kg, respectively. For quality control, blanks control, sample replicates (20%), and standard reference materials (GSS-5/GBW07405) were included in each batch of sample digestion and chemical analysis. And the relative standard deviations were less than 5%.

### 2.3. Methodology

#### 2.3.1. Exploratory Analysis

Identification of soil heavy metal sources on the regional scale (i.e., up to 1000 km^2^, almost the area of a county) using receptor models is difficult because of the heterogeneity of the parent materials of the soil and the high variability of anthropogenic activities. Thus, in this study, the entire region was partitioned using the spatial distribution of the pollution sources based on the enterprise land survey and the collection of data on the environmental factors. Moreover, this idea could help deal with the rotation ambiguity (i.e., different results obtained via PMF may generate similar model fitting) by decreasing the number of columns in the factor contribution matrix and the number of rows in the factor profile matrix.

After this, correlation analysis was conducted on the concentrations of the elements in the samples and the environmental factors. Because of the heavy rainfall in the study area, surface runoff and soil water flow must affect the spatial distribution of soil heavy metals, and water flow is closely related to topographic characteristics. Elevation (EL) directly reflects topographic features, while humidity index (HI) quantifies topographic control over basic hydrological processes. EL and HI were obtained via digital elevation model (DEM) data processing. DEM data came from a geospatial data cloud platform (https://www.gscloud.cn/, accessed on 8 July 2021), with a resolution of 30 m. The ELs of the samples were obtained directly using a spatial overlay, while the HI was determined from the topographic HI [23]:(1)$${\mathrm{HI}}_{i}=\ln\left(a_{i}/tan\beta_{i}\right),$$
where ${\mathrm{HI}}_{i}$ represents the humidity index at surface point i; $a_{i}$ represents the specific catchment area, i.e., the contributing upslope area per unit width of the contour; and $\beta_{i}$ is the gradient at point *i*.

The other environmental factors considered were distance (Dist) and direction (Dir) from the nearest pollution source, which were estimated using the Near tool in ArcGIS version 10.4.1 (Esri, Redlands, CA, USA).

#### 2.3.2. Source Apportionment via PMF Model

Similar to principal component analysis, PMF is a typical analytical factoring technique, which is based on the study of Paatero and Tapper [24,25]. In this study, the PMF 5.0 program was used to conduct the source apportionment of the soil heavy metals. The receptor sites were defined as the matrix relationship of a two-dimensional factor analytic with a residue matrix,
(2)X=GF+E,
or in the component form,
(3)$$x_{ij}=\overset{K}{\underset{k=1}{\sum }}g_{ik}f_{kj}+e_{ij},$$
where $x_{ij}$ represents the measured sample concentration, $g_{ik}$ represents the contribution of the kth factor for the ith sample, $f_{kj}$ represents the composition of the jth element within the kth factor, and $e_{ij}$ is the residual error.

Matrix G and matrix F were approximated using the PMF model to minimize the objective function Q under the constraint of non-negative contributions, which relies on more physically significant assumptions than other factor analysis methods [25].
(4)$$Q=\overset{n}{\underset{i=1}{\sum }}\overset{m}{\underset{j=1}{\sum }}{\left(\frac{e_{ij}}{u_{ij}}\right)}^{2},$$
where $u_{ij}$ represents the uncertainty of the jth chemical element for sample i.

#### 2.3.3. Spatial Pattern Analysis

Once contents of the heavy metals, which originated from industries of concern, were confirmed approximately through the above-described processes, these data were used to estimate the heavy metals of critical concern on a 200 m interval grid across the study area. Multivariable CK was chosen because homologous heavy metals could help each other to improve the accuracy through this method [26,27], and the estimation function was as follows:(5)$$Z^{*}\left(x\right)=\overset{n}{\underset{i=1}{\sum }}Z\left(x_{i}\right)\Gamma_{i},$$
where $x_{1},\;\cdots ,x_{n}$ represent the locations of the samples, and $Z_{1}\left(x\right),\;\cdots ,\;Z_{m}\left(x\right)$ represent the values of the multivariates at location x. $\Gamma_{i}$ represents the weighted vector. To determine $\Gamma_{i}$, an unbiased estimation was made with the smallest variance of error. Furthermore, by introducing Lagrangian multipliers and determining the derivation of $\Gamma_{i}$, linear equations were derived with semi-variogram and cross-variogram functions [28]. The semi-variogram was obtained by fitting the models, including Matérn, spherical, exponential, and power function models, while the cross-variogram functions were calculated as follows:(6)$$\gamma_{ij}\left(h\right)=\frac{1}{2}\left[\gamma_{ij}^{+}\left(h\right)-\gamma_{ii}\left(h\right)-\gamma_{jj}\left(h\right)\right],$$
where $\gamma_{ii}\left(h\right)$ represents semi-variogram of the ith variate $Z_{i}\left(x\right)$, $\gamma_{ij}\left(h\right)$ represents the cross-variogram between the ith and jth variates, and $\gamma_{ij}^{+}\left(h\right)$ represents the semi-variogram of $Z_{ij}^{+}\left(x\right)$, which is equivalent to $Z_{i}\left(x\right)+Z_{j}\left(x\right)$.

It must be emphasized that the auxiliary variables used in this study not only have a significant correlation, but they also have the same origin as the heavy metals from the mining and smelting sources, which increases the accuracy of the spatial prediction and provides a good foundation for the subsequent analysis using the environmental factors. In order to measure the error, the method of Zhang and Wang (2009) [29] and cross-validation were used depending on the predictions and variances of the predictions derived from the remaining observations. The predictive accuracy scores were calculated as follows:(7)$${𝓏}_{i}=\frac{Z\left(x_{i}\right)-{\hat{Z}}_{-i}\left(x_{i}\right)}{\sigma_{-i}},$$
where $Z\left(x_{i}\right)$ represents the observation made at location $x_{i}$ for i=1,⋯,n; and ${\hat{Z}}_{-i}\left(x_{i}\right)$ is the drop-one prediction based on all of the data $Z\left(x_{j}\right)$ for j≠i. $\sigma_{-i}$ is the corresponding standard deviation. Then, the mean of ${𝓏}_{i}$ becomes the z-score for the first index for evaluating the error conditions. Another predictive score is the root mean square error (RMSE):(8)$$\mathrm{RMSE}={\left[\frac{1}{n}\overset{n}{\underset{i-1}{\sum }}{\left(Z\left(x_{i}\right)-{\hat{Z}}_{-i}\left(x_{i}\right)\right)}^{2}\right]}^{1/2},$$

The RMSE should be as low as possible value, while the z-score should be close to 0.

Then, in this study, the heavy metals from the industries of concern were analyzed based on the environmental factor, i.e., the distances to the nearest pollution sources. The LWR method was used [30], and the target equation was minimized as follows:(9)$$\sum \omega^{\left(i\right)}{\left(y^{i}-\theta^{T}x^{\left(i\right)}\right)}^{2},$$
where $\omega^{\left(i\right)}=exp\left(-\frac{{\left(x^{\left(i\right)}-x\right)}^{2}}{2\tau^{2}}\right)$, which is the weight based on the degree of proximity to the predicted point. If the value of $\left|x^{\left(i\right)}-x\right|$ is very small, $\omega^{\left(i\right)}\approx 1$; while for a very large value, $\omega^{\left(i\right)}\approx 0$. τ indicates the rate of decrease with the degree of proximity. Finally, this regression method was used to analyze the variations in the heavy metal contents with distance from the nearest source, and two-dimensional fitting lines for the 95% confidence interval were obtained.

## 3. Results

### 3.1. Descriptive Statistic and Analysis of Variance Analysis of Samples

The descriptive statistics of the topsoil heavy metals in the two subregions are presented in Table 1. Higher heavy metal concentrations were observed in R1, including higher As (108.61%), Ni (123.47%), Cr (114.66%), Sb (164.03%), and Cu (116.80%) levels than those in R2; while the Cd, Pb, and Zn concentrations in the two subregions were similar. The coefficients of variation (CVs) of the two regions for As were significantly different than those of the other heavy metals, which may indicate that As and Cu concentrations in the two subregions have different spatial variabilities. Furthermore, the CVs of Cd, Pb, and Sb were all high (>50%), indicating that extrinsic factors strongly affected the enrichment of these heavy metals. Additionally, with the exceptions of Sb, Cr, and Ni, soil heavy metals of the different land use greatly surpassed risk screening values [31,32], indicating that As, Cd, Pb, Zn, and Cu may pose a threat to human and plants (Appendix A). Specifically, 92.32%, 100%, 80.12%, 62.54%, and 82.67% of the dry land soil samples surpassed the risk screening values for As, Cd, Pb, Zn, and Cu. For paddy land, As (97.14%), Cd (100%), Pb (72.14%), Zn (45.23%), and Cu (87.55%) exceeded their corresponding risk screening values. In garden plot samples, As (82.22%), Cd (100%), Pb (74.58%), Zn (49.34%), and Cu (70.52%) exceeded risk screening values.

An overview of the exploratory analysis dealing with the correlations between the elements and the environmental factors in R1 and R2 are presented in Appendix A. The Pearson’s correlation coefficients between the soil heavy metals, major elements (K, Ca, Mn, and Fe), and environmental factors (Dist, HI, and EL) for the samples from the two subregions were calculated (Appendix A). In R1, the correlations between As, Cd, Pb, Zn, and Cu were all significant (*p* < 0.01) moderately or strong positive (Appendix A), and Sb was moderately correlated with As and Cd. Dist exhibited low negative correlations with Pb, Zn, and Cu. The other correlations between the environmental factors and soil heavy metals were negligible. In R2, Pb was strongly positively correlated with Zn, while As was strongly positively correlated with Cd, Cu, and Sb. Similarly, Dist was slightly negatively correlated with Pb, Zn, and Cd, while the other correlations were negligible. In both R1 and R2, the major elements K, Mn, and Fe all exhibited at least moderate correlations, but Ca performed differently, which may be due to the different geological environments. Next, analysis of variance (ANOVA) [33] was applied to examine the differences in the soil heavy metal contents in different directions from the nearest pollution sources (Appendix A). The ANOVA results revealed that the number of differences between the means (*p* < 0.05) was always larger toward the south. For example, in R1, all of the means of the heavy metal contents in the samples whose nearest sources were located to the south were larger than those whose nearest sources were located to the west, and significant differences were identified for six heavy metals. Generally, samples with pollution sources located to the south always had positive significant differences compared with the other directions.

### 3.2. Source Identification

The receptor data for the two subregions, including eight heavy metals and four major elements, were used as the input data for the PMF. Originally, various trials with different numbers of factors were conducted, and the results revealed that a small number of factors (i.e., three or four factors) resulted in poor fitting, while six factors were excessive because the samples were mainly located within reach of the pollution sources, which was also indicated by the loss function’s Q value. Thus, the results for the two subregions using five factors both provided a reasonable interpretation with good fitting (i.e., R2 for heavy metals ≥0.50) and consistency with the field survey. As was previously mentioned, the model dealt with the rotational ambiguity by exploring different values of the rotational parameter Fpeak (between −1 and +1, step = 0.1), and −0.5 was adopted for R1, while no change was adopted for R2. The contributions are presented in Figure 2.

#### 3.2.1. Factors Related to the Industries of Particular Concern

The industries of particular concern in this study were nonferrous metal mining and smelting, which were also the focus of the field survey. The factors were categorized into two groups: related to industries of concern and others (Figure 2). The main categorization rule was the pollution characteristics of the soil heavy metals, including As, Cd, Cu, Pb, Zn, and Sb, which have been reported in previous studies [19,20,21]. In R1, factor 1 (R1F1) mainly explained the Pb (69.55%) and Zn (44.02%), and it also made moderate contributions (approximately 20%) to Cr, Ni, As, Cd, and Cu, indicating that it may be a key contamination source for the study area. Factor 2 (R1F2) exhibited the same characteristics and made large contributions to Cd (61.92%), As (47.82%), Cu (36.77%), and Sb (27.92%). Furthermore, R1F2 was determined to be robust because it was almost unaffected by the rotations in many trials. In R2, R2F1 predominantly contributed to the Pb (41.84%) and Zn (39.46%), but it also contributed to the (24.91%) and As (22.12%). R2F2 contributed to the As, Cd, Pb, Zn, and Ni, with at least 28% contributions for each metal, which demonstrates that it was very likely a contamination source related to smelting. Therefore, the above factors contributed greatly to the As, Cd, Cu, Pb, Zn, and Sb, which were the main pollutants in the study area. The specific judgment will be discussed in the next section.

#### 3.2.2. Other Factors

First of all, R1F3 and R2F3 deserve more attention because they are both very conspicuous, with large contributions to the major elements (K, Mn, Fe, and Ca). It is almost certain that R1F3 and R2F3 come from the weathering of local minerals. They also make moderate contributions to some of the heavy metals due to the high background values in the study area. In addition, some of the samples were collected from farmland and gardens, so R1F4 and R2F4 could be related to agricultural activity. The chemical plants, which were part of the industry chains of the local mining and smelting, may be related to R1F5 and R2F5.

### 3.3. Spatial Pattern of Key Heavy Metals Related to Industries of Concern

In this section, only the heavy metals originating from factors related to the industries of particular concern, i.e., nonferrous industry, are discussed. Based on their contents, the spatial distributions of the affected areas were estimated on a 200 m interval grid. The CK method was used to build models for auto-variograms and cross-variograms, which were found to have exponential forms in this study (Appendix A). Then, based on the spatial distributions of the heavy metals (Figure 3), the variation trends of the heavy metals with distance from the nearest source were analyzed, and two-dimensional fitting lines for the 95% confidence interval were obtained (Figure 4).

In order to determine the effective auxiliary variables for CK analysis of each heavy metal, the error conditions were tested through numerous trials (Table 2), and the variogram models were constructed to identify the homogeneous characteristics (Appendix A). As can be seen from Table 2, all RMSEs of the CK results are less than those of Universal Kriging (UK). The z-scores of the CK results are closer to 0 except for the CK for Cu. Figure 3 shows that the heavy metal concentrations around the study area all severely exceed the background levels (Appendix A) due to the presence of nonferrous industry enterprises. However, the spatial distribution patterns of these elements were different. Before estimating the spatial distribution, the area with a small sampling density was excluded. The results revealed that the spatial patterns of the effects of the pollution sources were different in R1 and R2. In R1, the soil Cu and Sb contents were high, while the As, Cd, Pb, and Zn all exhibited spatial patterns consistent with the pollution sources in the two subregions. Soil As pollution was high in the entire area, and the high-value areas were consistent with the source distribution, especially in the region with clusters of smelters in the north. The excess degree of Cd was the highest, and the value in the highest value area exceeds the background value by more than 40 times. Heavy Pb and Zn pollution were observed in both R1 and R2, with values approximately 10 times higher than the background values. Heavy Cu and Sb pollution were observed in R1 but were not detected in R2.

Figure 4 shows the variations in the heavy metals with distance from the nearest source, and the contents decrease with increasing distance. In Figure 4a, the trend flattens from 1800 m to 2500 m. However, the decreasing trends of the other heavy metals did not flatten, and the changing nodes are denoted by red crosses in Figure 4. When the distance is greater than the distance of the change node, the heavy metal contents decreased more slowly, and the uncertainty increased. The changing nodes of As and Cd were both located at approximately 1800 m, while those of Cu, Pb, Zn, and Sb were all located between 1000 m and 1500 m.

## 4. Discussion

This integrated approach provides a method of apportioning the contributions of industries of concern to soil heavy metals and of estimating the spatial patterns. The information obtained can be used to develop suggestions for the prevention and control of pollution around polluted industrial land. Several studies have been performed on source apportionment and geostatistics [34,35,36], focusing on clarifying the sources. However, this study emphasizes the characteristics of the soil pollution around industries of concern by extracting certain factors via PMF and estimating the spatial distributions using CK with the homology of the elements.

In the field survey and soil sampling, the object of the study is the soil within a distance of 5 km around the industries of concern. With regard to the public information about the protective distance of the soil environment from polluting enterprises [37], it is suggested that the scope of influence of nonferrous industries is within 5 km, and some studies [38,39] have also collected samples within 4 km or 5 km. Through exploratory analysis, it was found that the soil pollution to the north of the pollution sources is more serious, which can be explained by the fact that the dominant wind is from the south. In addition, the Dist exhibited a low correlation with Pb, Zn, and Cu, and the correlation with the other heavy metals is negligible. Therefore, further research is needed to determine the trend based on a receptor model and regression analysis.

Apportionment of soil pollution sources, which clarifies all of the potential sources, requires uniform sampling of the entire area [40,41,42], but in this study, only the sources related to mining and smelting are considered. Therefore, we provide a brief discussion of the other factors, which mainly depend on the characteristics of the major elements and the land-use types of the sampling locations. As was previously stated, the main migration pathways in R1 were likely dry and wet deposition from mining and concentration and surface runoff from the tailings. Furthermore, the pollution related to the tailings left over from historical mining is very serious, and leaching tests in the study area have demonstrated that the pollution derived from the tailings contains large amounts of As, Zn, and Pb. Therefore, R1F1 may be associated with the tailings. R1F2 made considerable contributions to all of the heavy metals, so it may be related to dry and wet deposition from mining and concentration. In R2, R2F1 was dominated by Pb and Zn, so it clearly originated from the emissions from smelters based on the survey. In addition, R2F1 is also associated with the accumulation of Cd and As, which is attributed to the atmospheric deposition of emissions from smelters. This conclusion is consistent with the results of previous studies [43,44]. In addition, the receptor model can be regarded as a tool for extracting factors of concern, which helps to illustrate the variation trends of the spatial patterns described below.

While some reports provided the spatial patterns based on the factors’ contributions, in this study, the contents of heavy metals related to industries of concern were extracted based on a receptor model, and the similar characteristics of some of the heavy metals (e.g., As and Cd have similar profiles related to the nonferrous industry) were taken advantage of to improve the accuracy of the spatial estimation through CK. The spatial variation trends of the heavy metals around polluting enterprises were studied [38,45,46], and the problem of contingency cannot be avoided. Based on statistical analysis, when the variation in the studied objects is large, the more samples are used, and the clearer the trend will be. The spatial patterns obtained via PMF and CK exhibit clear trends in the heavy metal contents with distance from the nearest sources. Then, the trends flattened, or change nodes of the trendlines occurred due to weakening of certain sources’ influences as the homologous nature of the other sources increased. In addition, the uncertainty increased significantly, which may be due to the weakened influences or the decreased sampling density. The above data show that 1800 m may be a satisfactory pollution protective distance in the study area. The LWR method has the ability to process a large amount of data, so it is considered to be a better tool based on trials using many regression methods.

## 5. Conclusions

In this study, geostatistical and chemometric methods were combined to identify pollution sources and estimate the spatial patterns of the soil heavy metals around industries of concern. It was found that at the county level, subregions should be defined before the apportionment of the sources due to the large spatial variations in the polluting enterprises. The main factors related to the industries of concern were extracted using the characteristics of the key heavy metals, which have been reported. The contents of the heavy metals from the main factors can help each other improve the estimation accuracy of the CK due to their homology. The trendlines of the variations with distance from the nearest sources can be used to determine a pollution protective distance around the industries of concern, but additional research is required.

## Figures and Tables

**Figure 1 ijerph-19-07421-f001:**
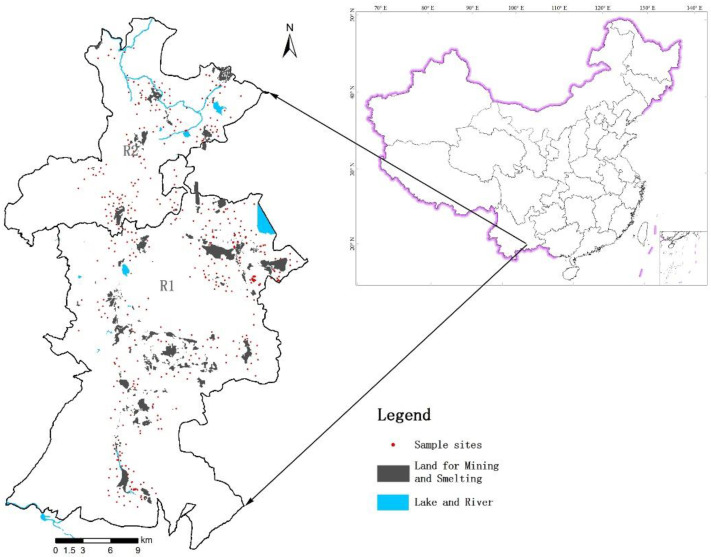
Land used for mining and smelting and the sampling sites in the study area.

**Figure 2 ijerph-19-07421-f002:**
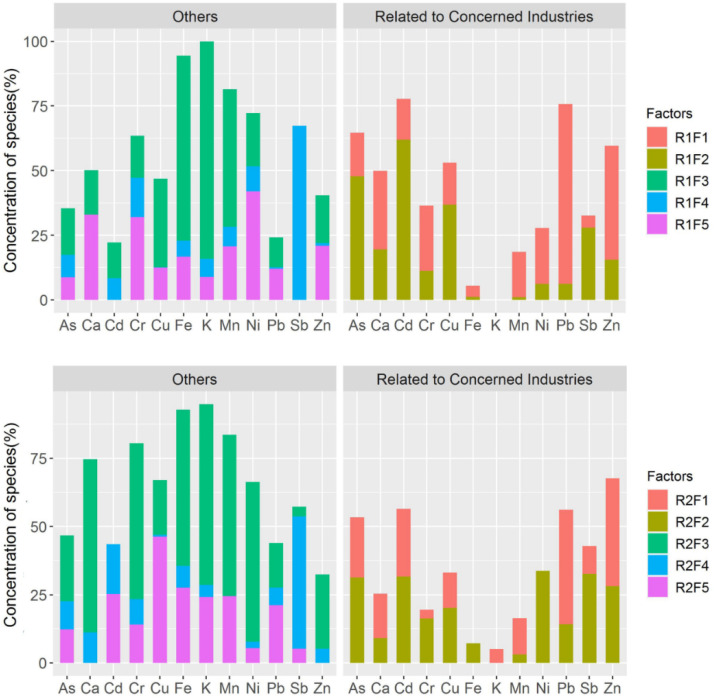
Contributions of the profiles for R1 (**top**) and R2 (**bottom**).

**Figure 3 ijerph-19-07421-f003:**
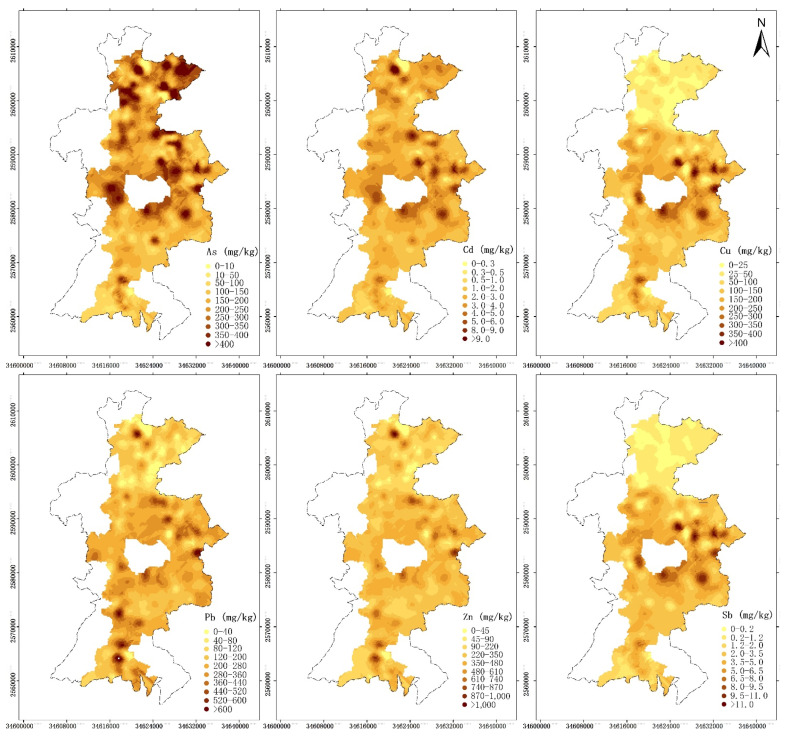
Spatial distributions of heavy metals obtained via PMF.

**Figure 4 ijerph-19-07421-f004:**
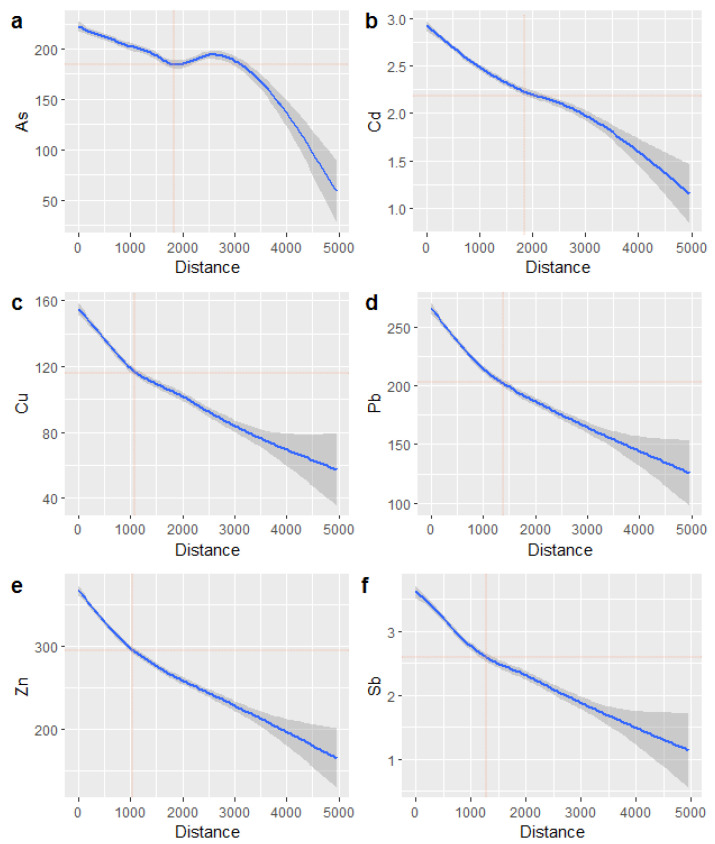
Variations in the contents of the heavy metals related to the industries of concern with distance from the nearest source. (**a**–**f**) The red crosses denote the changing nodes of the variation trends.

**Table 1 ijerph-19-07421-t001:** Descriptive statistics of the soil heavy metals.

Subregion	Element	Minimum	Maximum	Mean	SD	CV(%)
Subregion 1 (R1) (*n* = 389)	As (mg kg^−1^)	94.8	633.9	233.14	92.87	39.83
	Cd (mg kg^−1^)	1.33	11.24	2.77	1.4	50.51
	Pb (mg kg^−1^)	30.4	1236.4	322.46	183.5	56.91
	Zn (mg kg^−1^)	176.3	1134.5	503.72	139.06	27.61
	Ni (mg kg^−1^)	30.76	294.63	113.66	46.7	41.09
	Cr (mg kg^−1^)	14.86	463.4	154.57	70.68	45.73
	Sb (mg kg^−1^)	2.31	48.4	9.9	6.5	65.66
	Cu (mg kg^−1^)	119.42	897.05	277.29	119.62	43.14
	pH	5.42	8.06	6.52	1.3	19.94
Subregion 1 (R1) (*n* = 231)	As (mg kg^−1^)	18.8	538.2	214.65	131.41	61.22
	Cd (mg kg^−1^)	0.92	9.3	3.04	1.77	58.32
	Pb (mg kg^−1^)	161.34	1353.3	333.31	198.14	59.45
	Zn (mg kg^−1^)	348.7	1018.7	528.8	153.79	29.08
	Ni (mg kg^−1^)	32.41	220.9	92.05	40.58	44.08
	Cr (mg kg^−1^)	47.1	356.9	134.81	66.12	49.05
	Sb (mg kg^−1^)	0.27	19.18	6.04	5.08	84.19
	Cu (mg kg^−1^)	28.32	715.57	237.41	178.23	75.07
	pH	5.13	7.92	6.32	1.1	17.41

SD, standard deviation; CV, coefficient of variation.

**Table 2 ijerph-19-07421-t002:** Cross-validation results for cokriging and universal kriging.

Title 1	As		Cd		Cu	
entry 1	CK with Cd and Cu	UK	CK with As	UK	CK with Sb	UK
z-score	7.57 × 10^−4^	1.74 × 10^−3^	6.77 × 10^−3^	7.09 × 10^−3^	5.25 × 10^−3^	4.54 × 10^−3^
RMSE (mg/kg)	79.74	83.38	0.95	1.01	50.51	53.55
entry 2	Pb		Zn		Sb	
	CK with Zn	UK	CK with Pb	UK	UK with Cu	UK
z-score	1.05 × 10^−2^	1.08 × 10^−2^	1.30 × 10^−2^	1.35 × 10^−2^	−1.25 × 10^−3^	−1.46 × 10^−3^
RMSE (mg/kg)	90.42	102.19	113.18	127.22	1.39	1.48

## Data Availability

Data are contained within the article and Appendix A.

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
