# Peer review of "Apportionment and Spatial Pattern Analysis of Soil Heavy Metal Pollution Sources Related to Industries of Concern in a County in Southwestern China"

_ijerph, 2022, doi:10.3390/ijerph19127421_

Round 1

Reviewer 1 Report

  1. The authors should give the detailed informations for the soil samples, such as the representativeness, the land uses, the distance from the sampling sites to the industries, etc.
  2. Data statistical analysis should be added at the end of the Materials and methods.
  3. There were no any detailed data or informations for the environmental factors (Dist, HI, and EL) in the manuscript and also in the suplementary material.
  4. 1. Descriptive statistic and analysis of variance analysis of samples: The authors should clarify the land uses of the sampling sites? Agricultural land or development land? The assessment of heavy metals pollution should also based on the current soil environmental quality Risk control standard for soil contamination of agricultural land (GB 15618-2018) and development land (GB36600—2018)”.
  5. Table 1: The statistical results of soil pH should be added in this table.
  6. Line 266 and line 284: the “3.2.1. Factors related to the industries of particular concern” was repeated.
  7. 3.2. Source identification: The interpretations of the PMF results were rather general and vague. This part should be further improved and strengthened the source analysis and interpretations according to the local industrial and agricultural productions.
  8. Figure 3. Spatial distributions of heavy metals obtained via PMF: is it right? Please check and verify.

Author Response

Dear Reviewer:

Thank you for your comments concerning our manuscript entitled “Apportionment and spatial pattern analysis of soil heavy metal pollution sources related to industries of concern in a county in southwestern China”. The comments are all valuable and helpful for revising and improving our paper. We have studied comments carefully and have made correction which we hope meet with approval.

Reviewer 2 Report

I read the manuscript submitted for review with great interest.
The problem of soil pollution (by heavy metals and / or other) is ubiquitous and defining "areas of relevance" is extremely important.
It is clear that pollution is caused by the accumulation of materials over time and, in particular, in times when attention to public health was certainly not a priority.
Ultimately the manuscript is particularly interesting, well written and certainly worthy of attention.
I only have two small observations:
materials and methods, page 2 of 15, line 95: not semicolon after conducted.
resulta, page 6 of 15, lines 214-216: please delete because it is one of the phrases that are generally used as a guide to using the template.

Author Response

Dear Reviewer:

Thank you for your comments concerning our manuscript entitled “Apportionment and spatial pattern analysis of soil heavy metal pollution sources related to industries of concern in a county in southwestern China”. The comments are all valuable and helpful for revising and improving our paper. We have studied comments carefully and have made correction which we hope meet with approval. The responds to your comments are as flowing:

Point 1: materials and methods, page 2 of 15, line 95: not semicolon after conducted.
resulta, page 6 of 15, lines 214-216: please delete because it is one of the phrases that are generally used as a guide to using the template.

Response 1: The article has been modified according to your opinions. The semicolon in line 95 has been changed to comma. The first paragraph of section 3, which including lines 214-216, has been deleted.

Reviewer 3 Report

The paper is quite interesting even if the issue is well-studied and investigated worldwide. The approach is correct. I suggest to focus better the introduction, especially regarding the area, so the approach could be exported in other situations. Some parts of the section 2.1 could be moved in the introduction. The analytical part is quite poor, the authors have to give the analytical parameters (e.g., LODs) for ginning an idea of the analytical technique used. They should give more information on this issue. Why did not they analyze Hg, V? They should remake the table in alphabetic order, in this way they are confused, Check R2 (uppercase). There are differences between table 1 and figure 2, i.e., number of elements, why? Please, reduce section 2.3.2, it is too long the model description, this is a public health journal. Finally, check the English throughout the paper, in some it is obscure.

Author Response

Dear Reviewer:

Thank you for your comments concerning our manuscript entitled “Apportionment and spatial pattern analysis of soil heavy metal pollution sources related to industries of concern in a county in southwestern China”. The comments are all valuable and helpful for revising and improving our paper. We have studied comments carefully and have made correction which we hope meet with approval. The responds to your comments are as flowing:

Point 1: The approach is correct. I suggest to focus better the introduction, especially regarding the area, so the approach could be exported in other situations. Some parts of the section 2.1 could be moved in the introduction.

Response 1: The description of the study area was added in the introduction and the content of Section 2.1 was reduced accordingly.

Point 2: The analytical part is quite poor, the authors have to give the analytical parameters (e.g., LODs) for ginning an idea of the analytical technique used. They should give more information on this issue. Why did not they analyze Hg, V? They should remake the table in alphabetic order, in this way they are confused, Check R2 (uppercase).

Response 2: We have added LODs and quality control instructions to the second paragraph of section 2.2. The paragraphs read as follows: Upon receipt, the samples were dried in a lyophilizer and sieved (2 mm mesh), and then, the stones, litter, and roots were removed. The total major element contents (K, Ca, Mn, and Fe) of the samples were analyzed using an X-ray fluorescence spectrometer (Niton FXL analyzer, Thermo-Fisher Scientific, U.S.). Then, the samples were digested in HNO3 and H2O2 using method 3050B (USEPA, 1996). The total As concentration was analyzed using atomic fluorescence spectroscopy (AFS-9800, Haiguang Instrument Co., Beijing, China), and the Cd concentration was analyzed using graphite furnace atomic absorption spectrometry (contrAA700, Analytikjena, Germany). The other minor elements were measured using inductively coupled plasma optical emission spectrometry (Optima 5300DV, PerkinElmer, USA). The detection limits of As, Cd, Cu, Cr, Ni, Pb, Sb and Zn were 0.10, 0.05, 0.10, 0.10, 0.05, 0.10, 0.05 and 0.50 mg/kg. For quality control, blanks control, sample replicates (20%) and standard reference materials (GSS-5/GBW07405) were included in each batch of sample digestion and chemical analysis. And the relative standard deviations were less than 5%.

Point 3: There are differences between table 1 and figure 2, i.e., number of elements, why?

Response 3: Table 1 shows the statistics of soil heavy metals concerned in this paper, which is the object of this study. But Figure 2 shows all the elements used by the receptor model PMF, including heavy metals and major elements (K, Mn, Fe and Ca), which help identify the sources of soil heavy metals derived from weathering of local ores.

Point 4: Reduce section 2.3.2, it is too long the model description, this is a public health journal. Finally, check the English throughout the paper, in some it is obscure.

Response 4: In section 2.3.2, only the calculation principle of PMF and the condition of how to achieve the optimal solution are left, and the rest of the content is deleted. And We have further perfected English grammar.

Round 2

Reviewer 1 Report

The manuscript has been revised according to the reviews.

Author Response

(The authors gave the same response as above.)

Round 3

Reviewer 1 Report

The authors have revised the manuscript according to the reveiwers' comments. I recommend to accept it.

Author Response

Dear Reviewer:

Thank you for your comments concerning our manuscript entitled “Apportionment and spatial pattern analysis of soil heavy metal pollution sources related to industries of concern in a county in southwestern China”. The comments are all valuable and helpful for revising and improving our paper. We have studied comments carefully and have made correction which we hope meet with approval. English grammar and some details have been modified again this time. You can see the uploaded paper manuscript.